# Tactile roughness perception in the presence of olfactory and trigeminal stimulants

Lara A. Koijck[1], Alexander Toet[1] and Jan B.F. Van Erp[1,2]

[1] TNO, Soesterberg, The Netherlands
[2] Human Media Interaction, University of Twente, Enschede, The Netherlands

## ABSTRACT

Previous research has shown that odorants consistently evoke associations with textures and their tactile properties like smoothness and roughness. Also, it has been observed that olfaction can modulate tactile perception. We therefore hypothesized that tactile roughness perception may be biased towards the somatosensory connotation of an ambient odorant. We performed two experiments to test this hypothesis. In the first experiment, we investigated the influence of ambient chemosensory stimuli with different roughness connotations on tactile roughness perception. In addition to a pleasant odor with a connotation of softness (PEA), we also included a trigeminal stimulant with a rough, sharp or prickly connotation (Ethanol). We expected that—compared to a No-odorant control condition—tactile texture perception would be biased towards smoothness in the presence of PEA and towards roughness in the presence of Ethanol. However, our results show no significant interaction between chemosensory stimulation and perceived tactile surface roughness. It could be argued that ambient odors may be less effective in stimulating crossmodal associations, since they are by definition extraneous to the tactile stimuli. In an attempt to optimize the conditions for sensory integration, we therefore performed a second experiment in which the olfactory and tactile stimuli were presented in synchrony and in close spatial proximity. In addition, we included pleasant (Lemon) and unpleasant (Indole) odorants that are known to have the ability to affect tactile perception. We expected that tactile stimuli would be perceived as less rough when simultaneously presented with Lemon or PEA (both associated with softness) than when presented with Ethanol or Indole (odors that can be associated with roughness). Again, we found no significant main effect of chemosensory condition on perceived tactile roughness. We discuss the limitations of this study and we present suggestions for future research.

Corresponding author
Alexander Toet, lex.toet@tno.nl

## INTRODUCTION

When touching an object we perceive its texture not only through cutaneous and thermal input but also by using kinesthetic, auditory, and visual cues (*Lederman, 1982*). A growing body of research shows that the information processed in one sensory modality is modulated by the simultaneous activation of other sensory modalities

**Peer**J

(see *Driver & Noesselt, 2008*, for a review). As a result, tactile texture perception can for instance be influenced by audition (e.g., *Guest et al., 2002*; *Jousmäki & Hari, 1998*; *Klatzky & Lederman, 2010*; *Lederman, 1979*; *Werner & Schiller, 1932*), vision (e.g., *Guest & Spence, 2003a*; *Guest & Spence, 2003b*; *Werner & Schiller, 1932*), and even olfactory perception (*Churchill et al., 2009*; *Croy, Angelo & Olausson, 2014*; *Demattè et al., 2006*; *Gonçalves et al., 2013*; *Kikuchi, Akita & Abe, 2013*).

The inter-modal interaction between touch and vision is, for example, shown by the fact that bimodal visual and tactile input results in superior roughness discrimination of abrasive papers (*Heller, 1982*), and that the visual assessment of textile roughness is less accurate in the presence of simultaneously presented incongruent tactile samples (*Guest & Spence, 2003b*). There is also substantial neuroimaging evidence that vision and touch are intimately connected (for reviews see *Amedi et al., 2005*; *Sathian, 2005*; *Sathian et al., 2011*). Tactile discrimination is to a certain degree mediated by the visual cortex (*Lacey, Campbell & Sathian, 2007*; *Prather, Votaw & Sathian, 2004*; *Sathian, 2005*; *Sathian et al., 2011*; *Sathian & Zangaladze, 2002*; *Zangaladze et al., 1999*). Visual imagery mediates and is essential for some tactile tasks (e.g., orientation discrimination: *Sathian & Zangaladze, 2002*; *Zangaladze et al., 1999*).

Evidence for crossmodal interactions between the tactile and auditory sensing modalities are the observations that people's perception of the roughness of abrasive papers (*Guest et al., 2002*), the crispness of potato chips (*Zampini & Spence, 2004*), or even the texture of their own hands (*Jousmäki & Hari, 1998*) can be modified simply by manipulating the frequency content of the touch-related sounds. Brain studies have shown that the processing of sound in the auditory cortex is modulated by the simultaneous presentation of a tactile stimulus (*Kayser et al., 2005*), while sound can activate subregions of the medial ventral stream most strongly associated with the visual processing of surface properties of objects (*Arnott et al., 2008*).

Several studies have shown that olfaction can also interact with tactile perception. For example, the perceived smoothness (*Demattè et al., 2006*) and textural quality (*Laird, 1932*) of odorized fabrics depends on their odor. Lip balm feels smoother with lemon scent than with vanilla scent (*Kikuchi, Akita & Abe, 2013*). The perceived greasiness and spreadability of cream and gel formulations is influenced by the presence and type of fragrance (*Gonçalves et al., 2013*). Shampoo fragrance affects the perceived texture of both product and hair (*Churchill et al., 2009*). Touch pleasantness decreases in the presence of an unpleasant odor (*Croy, Angelo & Olausson, 2014*). In addition, odors consistently evoke associations with textures and their tactile properties like smoothness and roughness (*Spector & Maurer, 2012*). For instance, odors acquire their somatosensory tactile-like qualities during tasting experiences (*Stevenson & Mahmut, 2011*).

The human nose detects volatile compounds via at least two sensory systems. The olfactory system detects chemicals using specialized receptor neurons distributed on a limited dorsal area of the nasal mucosa and sends signals to the brain via the first cranial (olfactory) nerve. In the nose, mouth, eyes, and other facial areas, the trigeminal system detects chemicals using the more widely distributed free endings of the fifth cranial

(trigeminal) nerve. The olfactory system is more dedicated to identification of the hedonic and alimentary aspects of an odorant, whereas the trigeminal system mediates protective functions and reflexes by signaling somatosensory warning signals like cooling, numbness, tingling, itching, burning and stinging. Both systems use overlapping pathways that interact at multiple levels (*Rombaux et al., 2013*).

Most odorants stimulate both the olfactory and the trigeminal system. Since activation of the trigeminal nerve can evoke haptic sensations it may be regarded as a kind of tactile sense (*Lundström, Boesveldt & Albrecht, 2011*). Thus, when our nose detects an odorant, we may smell it, feel it, or both. This suggests that olfactory stimulation of the trigeminal nerve could be associated (and thus interfere) with simultaneous tactile perception. In addition, it has also been shown that there are stable semantic crossmodal associations between odors and somatosensory attributes. Odors that are judged more irritating and unpleasant are typically categorized as rougher/grittier (*Stevenson, Rich & Russell, 2012*). Also, a masculine smell is typically associated with a rough texture while a feminine smell is seen as congruent with a smooth texture (*Krishna, Elder & Caldara, 2010*). It appears that crossmodal odor associations are automatically activated even without conscious odor perception (*Seigneuric et al., 2010*).

In this study, we investigate the influence of olfactory stimulation on the perception of tactile surface roughness. Previous studies that reported interaction effects between olfaction and tactile perception focused on the hedonic valence of the olfactory stimuli (*Croy, Angelo & Olausson, 2014*; *Demattè et al., 2006*). Unpleasant odors were found to bias tactile perception towards roughness (*Demattè et al., 2006*) and unpleasantness (*Croy, Angelo & Olausson, 2014*). Pleasant odors showed an effect when contrasted with unpleasant ones (*Demattè et al., 2006*) but did not induce a significant bias by their own (merely a tendency: *Croy, Angelo & Olausson, 2014*; *Demattè et al., 2006*). In the present study, we first investigated the influence of ambient chemosensory stimuli with different roughness connotations on tactile roughness perception (Experiment I). In addition to a pleasant odor with a connotation of softness, we also included a trigeminal stimulant with a rough, sharp or prickly connotation. However, it could be argued that ambient odors may be less effective in stimulating crossmodal associations since they are by definition extraneous to the tactile stimuli. We therefore performed a second experiment in which the olfactory and tactile stimuli were presented in synchrony and in close spatial proximity in an attempt to optimize the conditions for sensory integration (Experiment II). In addition, we included pleasant and unpleasant odorants that are known to have the ability to affect tactile perception. If olfaction does indeed modulate tactile roughness perception, we expect that the presence of an odorant may bias tactile roughness perception in the direction of its associated characteristics.

In addition to furthering our understanding of multisensory smell-touch interactions, the results of this study may be of interest for the development and evaluation of for instance cleaning products and cosmetics, which typically combine fresh or floral fragrances with trigeminal stimulation from substances such as solvents (e.g., alcohol).

# EXPERIMENT I: AMBIENT ODOR

In Experiment I, we measured perceived tactile surface roughness in the presence of two different ambient odorants: a floral odor with no trigeminal stimulation that is typically associated with softness and femininity, and a trigeminal odorant with a rough, sharp or prickly connotation. The odors were presented at near threshold levels since low salient scents are known to affect product evaluation more easily independent of the degree of congruency, probably because observers are less able to discount the effect of an odor when it is not noticed (*Bosmans, 2006*). We hypothesize that compared to a No-odorant (clean air) condition, (H1) tactile texture perception will be biased towards smoothness in the presence of the ambient odorant with a soft or smooth connotation, whereas (H2) tactile texture perception will be biased towards roughness in the presence of the ambient odorant with a rough, sharp or prickly connotation.

## Methods
### *Participants*

Twenty-four non-smoking participants (12 males, 12 females) ranging in age from 18 to 50 years (mean age 35 years) took part in the experiment. The sample size was estimated from a statistical power analysis, based on data from *Demattè et al. (2006)*. The effect size in this study was 0.6 which is considered to be medium (*Cohen, 1988*). With an alpha $= .05$ and power $= .90$ the projected sample size needed with this effects size (G*Power 3.1, *Faul et al., 2007*; *Faul et al., 2009*) is approximately $N = 24$.

The participants were recruited from the TNO database of volunteers. All participants reported having a normal sense of smell and touch, and no history of olfactory or somatosensory dysfunction. Since smokers are poorer at detecting phenyl ethyl alcohol (used as an olfactory stimulus in this study) than non-smokers (*Hayes & Jinks, 2012*), we used smoking as an exclusion criterion. All participants were naïve to the purpose of the experiment: they were only informed that the study was about roughness perception in the absence of vision and hearing. Participants were requested to refrain from using hand lotion or crème and from wearing scented body lotions or perfumes in the morning of the experiment, since skin hydration significantly affects tactile roughness perception (*Gerhardt et al., 2008*; *Verrillo et al., 1998*) and the presence of cosmetic perfumes might interfere with the odorants used in this study. The participants read and signed an informed consent prior to the experiment. The experimental protocol was reviewed and approved by the TNO Ethics Committee and was in accordance with the Helsinki Declaration of 1975, as revised in 2013 (*World Medical Association, 2013*). The participants received 25 Euros for participating in the experiment, which lasted about 1.5 h.

### *Apparatus and materials*

The tactile stimuli in this study were samples of sandpaper (3M$^{TM}$ WetorDry$^{TM}$ abrasive paper: see www.3M.com) with six different grades of roughness. The sandpaper grit value (i.e., the approximate amount of sharp particles per square inch) was adopted as a measure for tactile roughness. Lower grit values correspond to higher tactile roughness (*Heller, 1982*). The grit values of the samples used in this study were respectively 60, 80,

180, 280, 400 and 500 (similar to the range used in previous studies, e.g., *Guest et al., 2002*; *Heller, 1982*; *Jones & O'Neil, 1985*; *Rexroad & White, 1988*; *Stevens & Harris, 1962*; *Verrillo, Bolanowski & McGlone, 1999*). The samples were mounted in rectangular plastic frames with a size of $10 \times 15$ cm$^2$. A pilot study confirmed the results of previous studies that the different grades of sandpaper were indeed discriminable on their perceived roughness. The physical roughness of the six grades of sandpaper was verified by microscopic examination and the surface structure of the sandpaper samples was further assessed by the use of a surface analyzer (a Sensofar PLµ 2300 optical imaging profiler: http://www.sensofar.com).

During the experiments, the participants wore glasses that completely blocked their sight (the glasses were made opaque with black tape) and sound-attenuating earmuffs (BILSOM 717—700-Series, EN 352-1) which reduced the ambient sound by 23 dB. These measures served to eliminate any visual or auditory surface roughness cues and to ensure that the participants only received tactile and olfactory input when estimating the roughness of the sandpaper samples. The sound reduction by the earmuffs was such that that the participants were still able to communicate with the experimenter. The participants' hands were gloved by cotton work gloves, with the index finger of the glove on their preferred hand removed. In this way all participants were restricted to touching the stimuli with the tip of the index finger of their preferred hand.

The trigeminal stimulus was Ethanol (73.5% volume percentage, diluted with propyleneglycol or PG). The olfactory stimulus was phenyl ethyl alcohol (PEA, 25% volume percentage, diluted with PG). Ethanol is a largely trigeminal odorant that can cause nasal irritancy at values above the olfaction threshold (*Cometto-Muñiz & Cain, 1990*; *Mattes & DiMeglio, 2001*). In contrast, PEA is a substance with a rose-like odor which is only odiferous and has minimal intranasal trigeminal properties (*Brand & Jacquot, 2001*; *Cometto-Muñiz & Cain, 1990*; *Doty et al., 1978*), and which is generally considered pleasant (*Khan et al., 2007*). Rose-like odors like PEA are typically associated with softness and femininity (*Thiboud, 1994*), whereas Ethanol is often associated with roughness (*Demiglio & Pickering, 2008*; *Jones et al., 2008*). All chemical substances were obtained from Sigma-Aldrich (www.sigmaaldrich.com).

The measurements were performed in three separate experiment rooms of equal size ($3.5 \times 5.5 \times 2.8$ m$^3$) and temperature (20 °C), that were shielded from external noise. Each room contained a desk that was covered with a black opaque tablecloth which reached down to the floor. The test solutions were diffused in the rooms through commercial electronic dispensers (small Xenon electric scent diffusers: http://www.scentaustralia. com.au/products/scent-diffuser-xenon.html) that were placed out of sight underneath the desks. A tube led the air with the test solution from the diffuser in the direction of the participant through a small hole in the tablecloth. The tablecloths served as an extra precaution to prevent that the participants could see (even though they wore vision blocking glasses during the experiment) or touch the scent dispensers at any time. Because the earmuffs did not totally eliminate the sound from the diffusers, we recorded their sound and played it at the correct sound level from beneath the desk in the No-odorant

condition. This served to ensure that the background noise was similar in all three chemosensory conditions.

Each room was used to present a single odor condition (PEA, Ethanol or No-odorant). The No-odorant (clean air) condition served as a negative control for both the odor (PEA) and trigeminal irritation (Ethanol) conditions (*Smeets, Mauté & Dalton, 2002*).

Participants judged the perceived tactile roughness of the sandpaper samples using the method of absolute magnitude estimation (AME), a standard technique used in the study of subjective sensation magnitude (*Gescheider & Hughson, 1991*; *Verrillo, Bolanowski & McGlone, 1999*; *Zwislocki & Goodman, 1980*). AME requires participants to match their subjective impression of the size of a number to their impression of the intensity of a stimulus. The participants rated the roughness of the sandpapers on a scale that ranged from 1 (*least rough*) to 9 (*most rough*). The samples were renewed after every four participants to avoid any impairment of the sandpapers through extended touching.

Odor was intermittently diffused during the experiment (according to a 50% duty cycle with a period of one minute) so that the participants received fluctuating concentrations over time, thus preventing full adaptation. The perceived odor intensity should neither be overwhelming (to avoid eliciting inappropriate expectations in the participants: *Elmes & Lorig, 2008*; *Smeets & Dalton, 2005*; see also (*Loersch & Payne, 2011*; *Smeets & Dijksterhuis, 2014*)) nor too low (so that the odor stimulation would be ineffective). Ideally, odor intensity should be above the detection threshold but just beneath the awareness threshold. (The awareness threshold refers to a level of odor at an intensity that someone will only notice it if attention is paid to it.). A pilot experiment was performed to determine a setting of the dispensers and a duty cycle that resulted in a mean rating of 5 on a 9-point scale (from $1 =$ *not detectable* to $9 =$ *very intense*). The odor exposure level never exceeded 1,900 mg/m$^3$ (1,000 ppm, as determined with a MiniRAE 3000 photoionization detector, see www.raesystems.com) in accordance with the recommended limit for one hour exposure conditions as given by the Health Council of the Netherlands (*Dutch Expert Committee on Occupational Standards, 2006*). The room in which the test was performed was well ventilated prior to each session.

The instructions and the response scale which the participants could use to report their judgments were verbally explained by the experimenter at the start of the experiment. During the tasks the participants verbally reported their judgments, and the experimenter registered the responses on a response sheet.

### Experimental design and analysis

The experiment was performed according to a within-subjects repeated-measures design, with the independent variables odor (PEA, Ethanol and No-odorant), sandpaper roughness (grit values 80, 180, 280 and 400) and gender. The experiment consisted of three blocks of 48 trials (four trials of four sandpapers for each of the three chemosensory conditions). The presentation of the four sandpapers in the three chemosensory conditions was randomized, just as the order of presentation of the chemosensory conditions (blocks). A mixed design ANOVA was used to analyze the perceived roughness scores with gender as between-subjects and chemosensory condition and sandpaper roughness

as within-subjects independent variables. All statistical analyses were performed with IBM SPSS 20.0 for Windows (www.ibm.com). For all analyses a probability level of $p < .05$ was considered to be statistically significant.

### Procedure

After their arrival at the laboratory, the participants were welcomed in a central waiting room that was surrounded by the three experiment rooms. Here they first received a verbal introduction and instruction from the experimenter, after which they read and signed an informed consent form. Participants were informed that they would be repeatedly estimating the perceived tactile roughness of paper surfaces. The participants were then asked to put on the vision blocking glasses, earmuffs and gloves. The experimenter then guided them to one of the three experiment rooms. The participant and the experimenter both took place on opposite sides behind the desk. On each trial the experimenter placed a sandpaper sample on the table directly in front of the participant.

In each chemosensory condition the participants were first presented with the roughest sandpaper sample (grit value 60) and the smoothest sample (grit value 500), to enable them to build up a reference for the task ahead. No roughness ratings were given for these two samples.

When exploring the stimuli with their preferred hand, all participants were instructed to hold each panel by its edges, using the non-preferred, gloved hand. They estimated the magnitude of the perceived stimulus roughness by moving the uncovered index fingertip of their preferred gloved hand back and forth with a moderate force and velocity over approximately 4–6 cm of the sample surface. The participants were allowed to repeatedly examine a sample surface before indicating its roughness. The speed of hand movement was not controlled in this experiment, since perceived roughness is largely independent of scanning velocity when actively exploring a surface texture with the bare finger (*Lederman, 1983*; *Lederman, 1974*; *Yoshioka et al., 2011*).

In each chemosensory condition the participants rated the roughness of all four samples in a randomized order. Each sample was presented four times in four trials per chemosensory condition. Every 30 s the next sample was presented, to ensure that each participant spent the same amount of time in each chemosensory condition. A full run in each condition lasted 10 min.

After each block, the participants were led back to the waiting room for a 5-minutes break. During the break they removed their glasses and earmuffs and read a magazine. They could also drink some water if they wanted. The 5-minute break after each run served to minimize carry-over effects from one chemosensory condition to the next and to avoid reduced sensitivity through extended touching of the sandpapers. After the break the participant was guided to another room to perform the same task in another chemosensory condition. Hence, the participants performed exactly the same task in each chemosensory condition. After the third and final block, the participants were guided back to the waiting room where they removed their glasses and earmuffs. Then they filled out a demographic questionnaire and they were asked whether they had noticed anything particular in the environment during the three blocks. Finally, the participants

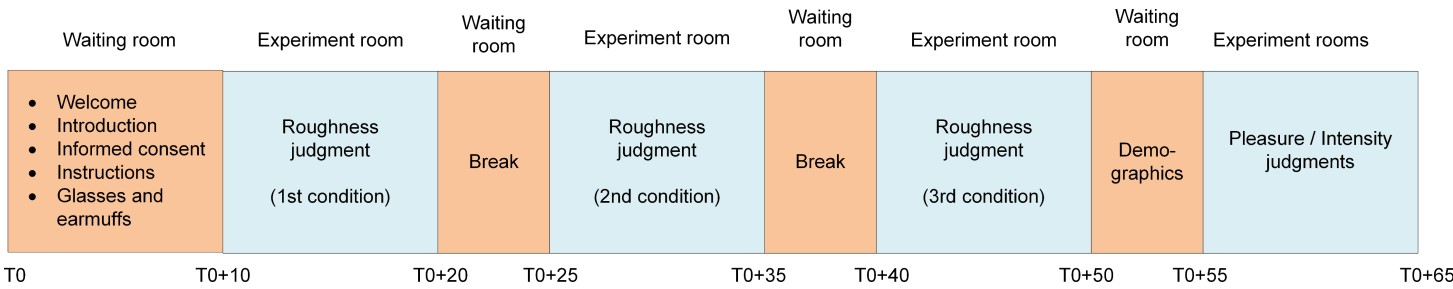

**Figure 1 Schematic representation of the procedure.**

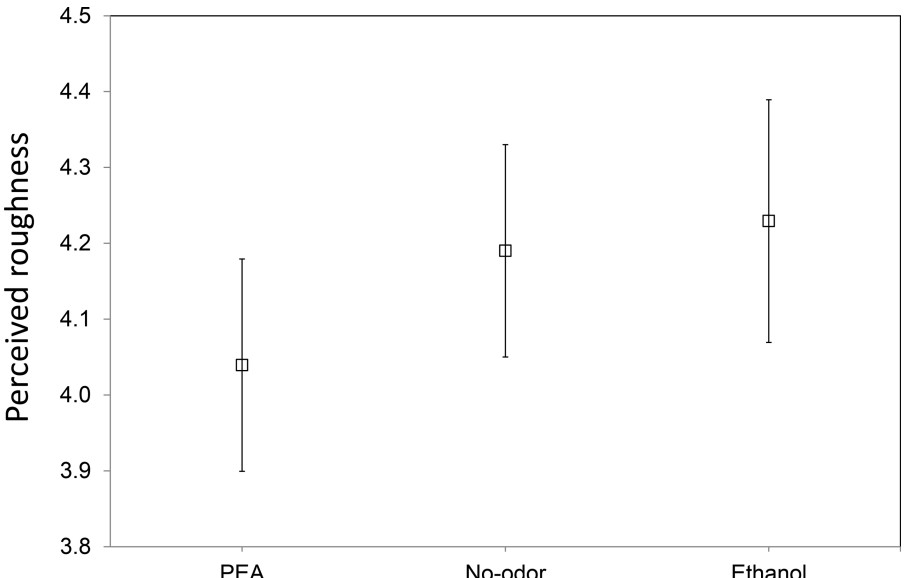

**Figure 2 Overall mean perceived roughness in the different ambient chemosensory conditions.** Mean perceived roughness over all four types of sandpapers (with grit values 80, 180, 280 and 400) for each of the three ambient chemosensory conditions (PEA, No-odorant, Ethanol), on a scale from 1 (*least rough*) to 9 (*most rough*).

were directed for the last time into each of the three experiment rooms (this time with their eyes and ears open) and they were asked to rate the intensity and pleasantness of the odor in each room on scales ranging from 1 (*not detectable/very unpleasant*) to 9 (*very intense/very pleasant*). See Fig. 1 for a schematic representation of the entire experiment.

## Results

Figure 2 shows the mean perceived roughness of the four sandpapers for each of the three chemosensory conditions (PEA, Ethanol, No-odorant). On first inspection the ranking of the mean roughness ratings in the three chemosensory conditions appears to agree with our expectations. Compared to the perceived roughness in the No-odorant condition ($M = 4.19$, $SE = 0.15$), participants judged the tactile surface roughness of the sandpaper samples higher when they were exposed to Ethanol ($M = 4.23$, $SE = 0.16$) and lower when they were exposed to PEA ($M = 4.04$, $SE = 0.15$). Further inspection of the data shows

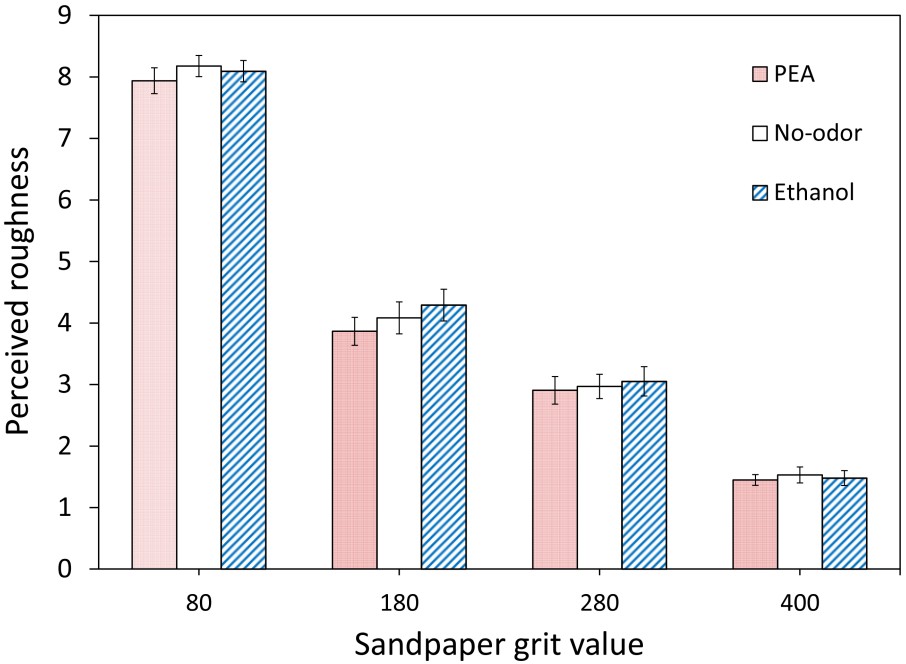

**Figure 3 Mean perceived roughness for each tactile stimulus in the different ambient chemosensory conditions.** Mean perceived roughness for each of the four types of sandpaper (with grit values 80, 180, 280 and 400) and each of the three ambient chemosensory conditions (PEA, Ethanol and No-odorant), on a scale from 1 (*least rough*) to 9 (*most rough*).

that the ranking of the mean roughness ratings in the three chemosensory conditions was in accordance with our expectations for the sandpapers with grit values 180 and 280, with the former having the largest difference in ratings (see Fig. 3). The difference between the mean roughness ratings in the three chemosensory conditions was minimal for sandpapers with grit values 80 and 400, and the ranking for these ratings did not agree with our expectations.

A mixed-design ANOVA with *gender*, *chemosensory condition* and *sandpaper roughness* as independent variables was conducted on the mean roughness ratings over the four repetitions. The results indicate that the roughness ratings did not differ between males and females: $F(1, 22) = 2.40$, $p = .14$; power with $\alpha$ set at .05 was .32. The results showed a significant main effect of sandpaper roughness, $F(3, 66) = 537.06$, $p < .001$. A post-hoc Turkey HSD test showed that participants were able to discriminate between all four sandpapers (all comparisons $p < .001$). There was no significant main effect of chemosensory condition, $F(2, 44) = 1.61$, $p = .21$. This suggests that chemosensory condition did not affect the roughness ratings for the four sandpapers. The results also show that there was no significant interaction between chemosensory condition and sandpaper roughness, $F < 1.0$. This reveals that the profile of ratings across sandpapers of different grit values was not different for the PEA, Ethanol and No-odorant conditions. Hence, both hypothesis H1 (people judge the tactile surface roughness of objects higher when they are exposed to a substance with a trigeminal component compared to a No-odorant or clean air condition) and hypothesis H2 (people judge the tactile surface

**Table 1 Mean perceived ambient chemosensory stimulus pleasantness and intensity.** Mean ratings (+ SE) of the perceived pleasantness (1 = *very unpleasant*, 9 = *very pleasant*) and intensity (1 = *not detectable*, 9 = *very intense*) for the three ambient chemosensory stimulus conditions in Experiment I.

| Chemosensory condition | Perceived pleasantness | Perceived intensity |
|---|---|---|
| PEA | 5.13 (0.52) | 7.04 (0.39) |
| Ethanol | 5.88 (0.35) | 4.96 (0.46) |
| No-odorant | 6.75 (0.36) | 2.29 (0.34) |

roughness of objects as lower when they are exposed to PEA, compared to a No-odorant or clean air condition) are not supported by our data.

The mean ratings on *pleasantness* and *intensity* of the three chemosensory conditions are listed in Table 1. All odorants differed significantly on their intensity and pleasantness, $p < .001$. The perceived pleasantness was near neutral (between 5.13 and 6.75) in all conditions. The perceived intensity varied from almost imperceptible (2.29) in the No-odorant condition, via near neutral (4.96) in the Ethanol condition to intermediate (7.04) in the PEA condition. The scent predominantly received floral labels in the PEA condition (16 out of 24). The Ethanol condition also evoked distinct associations in most participants (21 out of 24) ranging from *medicine* (7 out of 24) to *perfume* (2 out of 24), with some participants correctly reporting a scent of alcohol (7 out of 24). Most participants (14 out of 24) did not have any association in the No-odor condition, while some gave labels like *musty* (2 out of 24) or *nature* (5 out of 24). The fact that the participants consistently rated the intensity higher in the odorant conditions than in the No-odorant condition, predominantly reported a floral odor in the PEA condition, and reported appropriate associations in the Ethanol condition (i.e., substances that may contain alcohol like medicine and perfume), while no one reported noticing a smell during the experiments, suggests that the odorants were successfully administered at near-awareness threshold levels.

In principle, odors of different hedonic value may differentially affect perceived roughness. Therefore we explored the effects of PEA on roughness perception separately for likers and dislikers of PEA by conducting an independent samples t-test on the data of the PEA condition only with *rated roughness* as the dependent variable and *(dis)like PEA* as the grouping variable. Because the participants rated the pleasantness of the odors on a scale from 1 (*very unpleasant*) to 9 (*very pleasant*), we classified ratings 1–4 as unpleasant ($n = 10$ dislikers), rating 5 as indifferent ($n = 1$), and ratings 6–9 as pleasant ($n = 13$ likers). On average, participants rated the tactile surface roughness of the sandpapers in the PEA condition higher when they liked PEA ($M = 4.15$, $SE = 0.25$) than when they did not like PEA ($M = 3.94$, $SE = 0.19$). However, this difference was not significant, $t < 1.0$.

## Discussion

In Experiment I, we investigated the influence of ambient chemosensory stimuli with different roughness connotations on tactile roughness perception. To that end, we

measured the perceived tactile roughness of sandpapers with four different grades of surface roughness (grit values 80, 180, 280 and 400), in conditions with respectively clean air (control or No-odorant condition) and phenyl ethyl alcohol (PEA) and Ethanol as ambient odorants. We expected that compared to a No-odorant control condition, tactile texture perception would be biased towards (H1) smoothness in the presence of PEA since this odorant is typically associated with softness and femininity, and (H2) towards roughness in the presence of Ethanol since this odorant has a rough connotation due to its trigeminal nature. We found no significant main effect of chemosensory condition on perceived surface roughness. The results showed that there also was no significant interaction between chemosensory stimulation and sandpaper roughness. Thus, both our hypotheses (H1 and H2) were not confirmed.

Despite the lack of significance, the ranking of the mean rating responses on roughness in the three chemosensory conditions agreed with our expectations. The results revealed that the mean roughness ratings were higher in the Ethanol condition and lower in the PEA-condition compared to the No-odorant condition. Further analysis of the roughness ratings for each type of sandpaper individually showed that the ranking of the mean roughness ratings in the three chemosensory conditions was in accordance with our expectations only for the sandpapers with grit values 180 and 280, with the former having the largest difference in ratings. The variation in the roughness ratings for sandpapers with grit values 80 and 400 was minimal in the three chemosensory conditions and their ranking did not agree with our expectations. The consistency and small variation in the responses for the stimuli with the highest and lowest grit values may be due to the fact that the participants often recognized these stimuli from memory as being the extremes used in the actual test set and consistently gave them corresponding extreme ratings (9 or 1). This may be because the sandpapers with grit values of respectively 60 and 500 (the extremes used as anchors before the commence of the actual tests) were difficult to discriminate from sandpapers with the absolute extreme grit values of respectively 80 and 400. In the experiments, the participants may have changed their prior anchors (grit values 60 and 500) for the extremes of the subjective roughness scale (ratings 9 and 1) to the extreme grit values that actually occurred during a test (grit values 80 and 400) thereby automatically assigning them extreme ratings. In a debriefing after the experiment, we also asked the participants how much different types of sandpapers they thought they had rated during the actual tests. Most participants thought they had rated more than the 4 different types that were actually presented. In some tests, we presented the same sandpaper two times in a row, whereupon participants often answered with a different but comparable rating. This indicates that not all roughness ratings were based on memory.

An explanation for the lack of significance of the results from the Ethanol condition may be that the concentration to which the participants were exposed was too low to produce a noticeable physiological effect. For ethical reasons the ethanol concentration was limited in this study to the awareness threshold (*Health Council of the Netherlands, 2006*). As a result, the concentration in the room was below the irritation threshold so that most of the participants did not experience any negatively valenced chemosensory

effects. The participants rated the intensity of Ethanol as intermediate ($M = 4.96$ on a scale from $1 =$ not detectable to $9 =$ very intense). The scent received labels varying from medicine to perfume, with some participants correctly reporting a scent of alcohol. These findings suggest that we successfully administered Ethanol at a just noticeable level. However, nobody reported a prickling feeling. A lack of effect in the PEA condition does not contradict earlier reported findings. For instance, *Demattè et al. (2006)* showed that positively valenced odors do not bias roughness ratings but merely result in a tendency toward less rough ratings. Our results confirm this finding and show the same, non-significant trend.

## EXPERIMENT II: AMBIENT ODOR

In Experiment I, we found no effect of trigeminal and olfactory stimulation on tactile roughness perception. However, *Demattè et al. (2006)* observed significant differences between roughness ratings in odor conditions that were extremes on the dimension *pleasantness*. Their participants rated fabric swatches as feeling significantly softer when presented with a lemon (pleasant) odor than when presented with an unpleasant (animal-like) odor. In a related study, *Croy, Angelo & Olausson (2014)* recently found that an unpleasant (feces-like) odor reduces touch pleasantness. Considering this evidence, we performed a second experiment in which we included both lemon as an additional pleasant odor (in addition to PEA), and Indole as an additional unpleasant (feces-like) odor.

Research in multisensory perception and human neuroimaging studies have shown the importance of temporal and spatial congruence of incoming stimuli in establishing crossmodal associations (for a review see *Calvert, 2001*; *Calvert & Thesen, 2004*).

In Experiment I, we investigated tactile roughness perception in the presence of (continuously present) ambient odors administered at near-awareness threshold levels. It could be argued that the absence of an effect in Experiment I may be due to the fact that these conditions are sub- optimal for stimulating crossmodal associations. Note that we chose these conditions because they are characteristic for many ecological settings in which ambient odors from cleaning products, air refreshers, shampoos and cosmetics typically occur. However, since neither the temporal onset of an odor (e.g., *Frasnelli, Wohlgemuth & Hummel, 2006*; *Stevenson & Boakes, 2003*) nor the spatial location of its source are precisely coded in the olfactory system (e.g., *Kobal, Van Toller & Hummel, 1989*; *Porter et al., 2005*; *Spence et al., 2000*), strict spatio-temporal congruency may not be required to establish crossmodal associations. Previous studies indeed found that spatial proximity suffices (and strict spatial coincidence is not required) to establish tactile-olfactory interaction (*Croy, Angelo & Olausson, 2014*; *Demattè et al., 2006*). In an attempt to optimize the conditions for sensory integration we presented the olfactory and tactile stimuli in Experiment II in synchrony and in close spatial proximity.

Based on *Croy, Angelo & Olausson (2014)* findings we hypothesize (H3) that compared to a No-odor control condition, tactile stimuli will be perceived as rougher when simultaneously presented with an odorant (Indole) with an unpleasant (animal- or feces-like) quality. Based on the results of *Demattè et al. (2006)* we hypothesize that (H4)

tactile stimuli will be perceived as smoother when simultaneously presented with odorants that can be associated with softness (Lemon, PEA), than when presented with odors that can be associated with either roughness (Ethanol) or animals (Indole).

## Methods

### *Participants*

Thirty-six non-smoking participants (18 males, 18 females) ranging in age from 18 to 63 years (mean age 31 years) took part in the experiment. The participants were recruited by public announcements. All participants reported having a normal sense of smell and touch, and no history of olfactory or somatosensory dysfunction. All participants were naïve to the purpose of the experiment: they were only informed that the study was about roughness perception in the presence of smell. Participants were requested to refrain from using hand lotion or crème and from wearing scented body lotions or perfumes in the morning of the experiment. The participants read and signed an informed consent prior to the experiment. The experimental protocol was reviewed and approved by the TNO Ethics Committee and was in accordance with the Helsinki Declaration of 1975, as revised in 2013 (*World Medical Association, 2013*). The participants received 5 Euros for participating in the experiment, which lasted about 25 min.

### *Apparatus and materials*

The tactile stimuli in this experiment were the same as in Experiment I: samples of sandpaper (3M^TM WetorDry^TM abrasive paper; 3M, Saint Paul, Minnesota, USA) with six different grades of roughness (60, 80, 180, 280, 400 and 500), mounted in rectangular plastic frames with a size of $10 \times 15$ cm$^2$. The samples were renewed after every four participants to avoid any impairment of the sandpapers through extended touching.

The olfactory stimuli were phenyl ethyl alcohol (PEA) and Ethanol (both from Sigma Aldrich, Seelze, Germany), Lemon essential oil (De Tuinen, Amsterdam, Netherlands), Indole (De Hekserij, IJsselmuiden, Netherlands) and a No-odorant control condition. Indole is an aromatic heterocyclic organic compound with minimal trigeminal properties which is typically rated as unpleasant (*Bensafi et al., 2002*; *Grabenhorst et al., 2007*; *Grabenhorst, Rolls & Margot, 2011*; *Khan et al., 2007*). The odors were absorbed on a piece of cotton to ensure a better exchange with the air. The odor samples were kept and presented in odorless plastic flasks (60 ml, 3 cm in diameter at the opening). A pilot experiment (with 10 participants, 5 females) was performed to determine a concentration for each odor that resulted in a mean rating of 6 on a 9-point intensity scale (from 1 = *not detectable* to 9 = *very intense*).

To exclude any visual or auditory surface roughness cues, the participants wore glasses that blocked their sight and sound-attenuating earmuffs which reduced the ambient sound (see Experiment I). In addition, they wore cotton gloves with the index finger of the preferred hand removed so that they could only touch the stimuli with the tip of the index finger of their preferred hand. During the experiment, the participants were seated in a comfortable chair with their head supported by a chin rest. Olfactory stimuli were administered by placing the bottles containing the odorants on a rigid support that

was fixed to the chin rest at approximately 3 cm below the participant's nose. Subjects were requested to breathe normally during the experimental session. The experiment was performed in a small room that was well ventilated prior to each session to prevent any odor accumulation.

### Experimental design and analysis

The experiment was performed according to a within-participants repeated-measures design, with odor (PEA, Lemon, Ethanol, Indole and No-odorant), sandpaper roughness (grit values 80, 180, 280 and 400) and gender as independent variables. An experimental session consisted of four blocks of 20 trials (four trials of four sandpapers for each of the five chemosensory conditions). The combination of tactile and olfactory stimuli was randomized across trials, with the restriction that neither the same odor nor the same tactile roughness was presented on consecutive trials. All 20 (4 tactile samples × 5 odors) combinations occurred, and each combination was presented four times over the course of the experiment, resulting in a total of 80 trials. A mixed design ANOVA was used to analyze the perceived roughness scores with gender as between-subjects and chemosensory condition and sandpaper roughness as within-subjects independent variables. All statistical analyses were performed with IBM SPSS 20.0 for Windows (IBM, Armonk, New York, USA). For all analyses, a probability level of $p < .05$ was considered to be statistically significant.

### Procedure

After their arrival at the laboratory, the participants first received a verbal introduction and instruction from the experimenter, after which they read and signed an informed consent form. Participants were informed that they would be repeatedly estimating the perceived tactile roughness of paper surfaces while being exposed to different odors. The participant and the experimenter both took place on opposite sides of a desk. A chin rest was mounted on the desk in front of the participant. The participant was asked to adjust the height of the chair and the chin rest to a comfortable position. The participant then put on the opaque glasses, the sound-attenuating earmuffs and the gloves.

Then, the experimenter presented all five odorants in random order and asked the participant to rate both the intensity and pleasantness of the odorants on scales ranging from 1 (*not detectable/very unpleasant*) to 9 (*very intense/very pleasant*).

Next, the experimenter placed the two sandpaper samples with grit values 60 and 500 on the table directly in front of the participant. The participant was instructed to hold each panel in turn by its edges, using the non-preferred (gloved) hand, and to explore the stimuli by moving the uncovered index fingertip of the preferred (gloved) hand back and forth with a moderate force and velocity over approximately 4–6 cm of the sample surface. The participant was informed that these were respectively the roughest and smoothest samples that could be presented; this enabled the participant to build up a reference for the task ahead.

On each trial, the experimenter simultaneously placed a tactile sample on the table in front of the participant and a flask with an odor sample in the support below the

**Table 2 Mean perceived chemosensory stimulus pleasantness and intensity.** Mean ratings (+SE) of the perceived pleasantness (1 = very unpleasant, 9 = very pleasant) and intensity (1 = not detectable, 9 = very intense) for the five chemosensory stimuli in Experiment II.

| Chemosensory condition | Perceived pleasantness | Perceived intensity |
| --- | --- | --- |
| PEA | 6.47 (0.28) | 6.44 (0.19) |
| Ethanol | 4.11 (0.27) | 6.05 (0.29) |
| Lemon | 6.42 (0.25) | 6.83 (0.18) |
| Indole | 2.33 (0.20) | 7.19 (0.15) |
| No-odorant | 5.08 (0.11) | 1.64 (0.14) |

participant's nose, and said "YES" to inform the participant that a new sample was in position and ready for inspection. The participant verbally reported the classification rating (typically within a few seconds). The experimenter manually noted the rating on a response sheet and removed the (olfactory and tactile) stimuli. Every 20 s, a different tactile/olfactory stimulus combination was presented. The participant breathed plain room air during the intertrial intervals, which should serve to refresh their scent palette (*Grosofsky, Haupert & Versteeg, 2011*). A full run typically lasted about 25 min.

## Results

The *pleasantness* and *intensity* ratings of the five olfactory samples are listed in Table 2. All odorants differed significantly on their intensity and pleasantness, $p < .001$. PEA and Lemon were rated as pleasant, Ethanol as somewhat unpleasant, while Indole was rated as unpleasant. The No-odor control was rated near neutral. The perceived intensity of the four odorants varied between 6.05 (Ethanol) and 7.19 (Indole).

Figure 4 shows the mean perceived roughness of the four sandpapers for each of the five chemosensory conditions (PEA, Lemon, Ethanol, Indole and No-odorant). The ANOVA showed a significant effect of grit value: $F(3, 102) = 772.31$, $p < .001$ but no significant main effect of gender: $F(1, 34) = 0.00$, $p = .99$ (observed power .05) and of chemosensory condition: $F(4, 136) = 0.61$, $p = .66$ (observed power .20). Also, none of the interactions reached significance. A post-hoc Turkey HSD test on the main effect of grit value showed that participants were able to discriminate between all four sandpapers (all comparisons $p < .001$). Note that these effects replicate the results of Experiment 1.

Further inspection of the data reveals that the ranking of the mean roughness ratings was not in agreement with our hypothesis for any of the sandpaper grit values tested (see Fig. 5). The variation in mean roughness ratings between the five chemosensory conditions was minimal for sandpapers with grit values 80 and 400, while the ranking for two intermediate grit values was neither consistent nor in agreement with our expectations.

## Discussion

In Experiment II, we investigated the influence of olfactory stimuli with different hedonic (pleasant and unpleasant) and trigeminal (low like PEA or high like Ethanol) values on tactile roughness perception, with the olfactory and tactile stimuli presented in

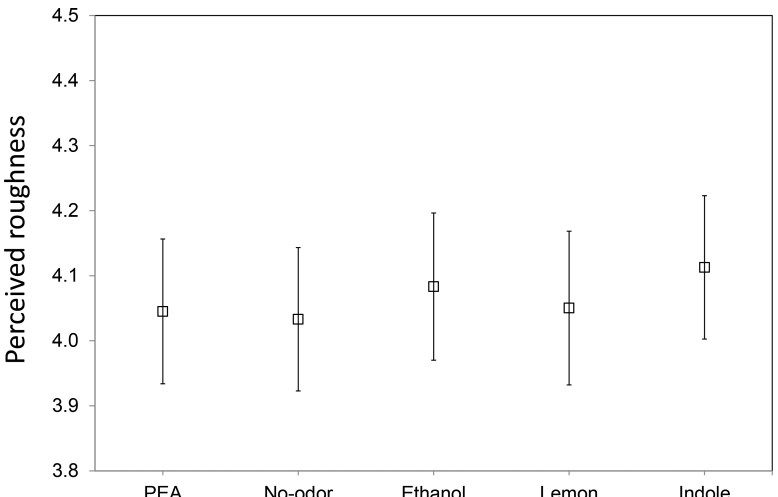

**Figure 4 Overall mean perceived roughness in the different synchronized chemosensory presentation conditions.** Mean perceived roughness over all four types of sandpapers (with grit values 80, 180, 280 and 400) for each of the five synchronized chemosensory presentation conditions (PEA, No-odorant, Ethanol, Lemon, Indole), on a scale from 1 (*least rough*) to 9 (*most rough*).

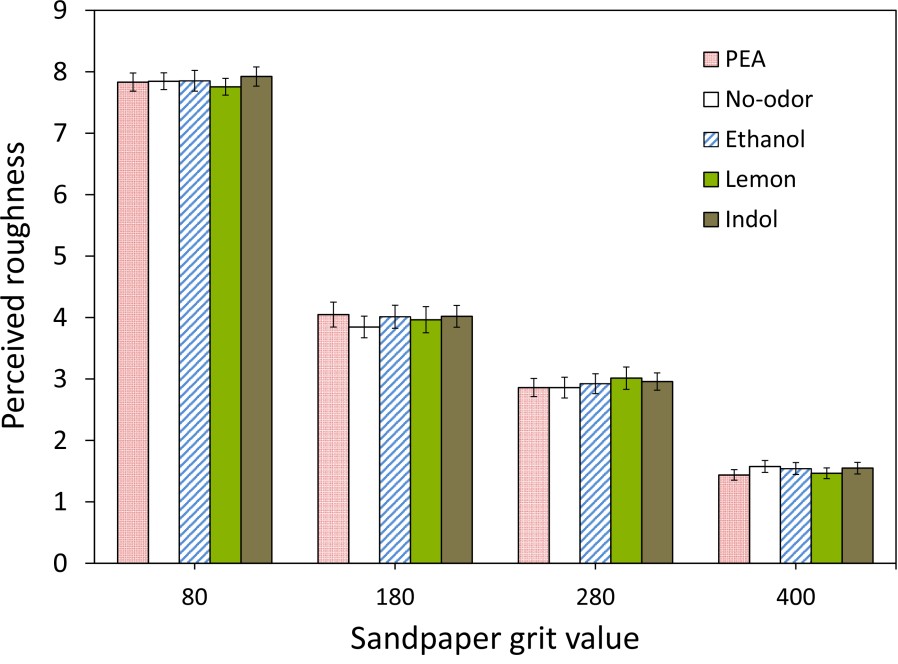

**Figure 5 Mean perceived roughness for each tactile stimulus in the different synchronized chemosensory presentation conditions.** Mean perceived roughness for each of the four types of sandpaper (with grit values 80, 180, 280 and 400) and each of the five synchronized chemosensory presentation conditions (PEA, No-odorant, Ethanol, Lemon, Indole), on a scale from 1 (*least rough*) to 9 (*most rough*).

temporal synchrony and in close spatial proximity. We also included pleasant (Lemon) and unpleasant (Indole) odors that are known to have the ability to affect tactile perception.

We expected (H3) that compared to a No-odor control condition, tactile stimuli would be perceived as rougher when simultaneously presented with Indole (which has an unpleasant quality). Also, we hypothesized that (H4) tactile stimuli would be perceived as smoother when simultaneously presented with Lemon or PEA (odorants that can be associated with softness) than when presented with Ethanol or Indole (odors that can be associated with roughness or animals).

As in Experiment I, we found no significant main effect of chemosensory condition on perceived tactile roughness. The results showed that there also was no significant interaction between chemosensory stimulation and sandpaper roughness. Thus, both our hypotheses (H3 and H4) were not confirmed.

## GENERAL DISCUSSION

In the two experiments reported here, we investigated the influence of olfactory and trigeminal stimulation on tactile roughness perception. Our results show no effect of olfactory or trigeminal stimulation on tactile roughness perception, independent of the hedonic valence or the trigeminality of the olfactory stimuli. The absence of an effect for pleasant odors is not surprising, given the fact that previous studies showed that pleasant odors by their own only show a weak tendency to induce a tactile bias (*Croy, Angelo & Olausson, 2014*; *Demattè et al., 2006*) and at best show a significant effect when contrasted with unpleasant odors (*Demattè et al., 2006*). The absence of an effect for unpleasant odors is somewhat unexpected, given the fact that unpleasant odors have previously been found to bias tactile perception towards roughness (*Demattè et al., 2006*) and unpleasantness (*Croy, Angelo & Olausson, 2014*).

In Experiment I, we used ambient odors at near-awareness threshold levels that were continuously present and by definition extraneous to the tactile stimuli. It could be argued that the resulting lack of spatio-temporal congruency may have prevented the establishment of crossmodal associations between the odors and the tactile stimuli. In Experiment II, we therefore presented the olfactory and tactile stimuli in synchrony and in close spatial proximity in an attempt to optimize the conditions for sensory integration. In addition, we included pleasant (Lemon) and unpleasant (Indole) odorants that are known to have the ability to affect tactile perception. However, the results of Experiment II again showed no effect of olfactory or trigeminal stimulation on tactile roughness perception. Although previous studies in multisensory perception and human neuroimaging have indeed demonstrated the importance of spatio-temporal stimulus congruence in establishing crossmodal associations (for a review see *Calvert, 2001*; *Calvert & Thesen, 2004*), other characteristics, like semantic congruence (*Krishna, Elder & Caldara, 2010*; *Stevenson, Rich & Russell, 2012*), are also known to play a crucial role in binding crossmodal associations. For instance, several studies have shown that ambient scent only significantly affects product evaluation when it is congruent with the targeted product (e.g., *Bosmans, 2006*; *Mitchell, Kahn & Knasko, 1995*), even when

the product itself has no inherent scent. For example, approach behaviors of shoppers for men's and women's clothing increased when a gender-congruent scent was present in the store (*Spangenberg et al., 2006*). This effect is stronger when the scent is not salient, since observers tend to discount the effect of an ambient scent when it is recognized as an extraneous stimulus, especially when the scent is incongruent (*Bosmans, 2006*). Hence, it seems that a significant interaction effect can only be expected between tactile stimuli and matching low-salient scents. Previous studies that did find crosssmodal interaction effects between olfaction and touch typically used naturalistic tactile stimuli like textile samples (*Demattè et al., 2006*; *Guest & Spence, 2003b*; *Laird, 1932*), cream and gels (*Gonçalves et al., 2013*; *Kikuchi, Akita & Abe, 2013*) or shampoo and hair (*Churchill et al., 2009*). These stimuli differed on two aspects from the sandpapers used in our experiments. First, they had associated scents or actively emitted scents that are probably more easily matched to their tactile profile. The lack of an effect in the present study may be due to the fact that sandpaper has no typical inherent smell and is not semantically congruent with any of the odors tested, except maybe with Ethanol, which has a rough, sharp or prickly connotation (*Demiglio & Pickering, 2008*; *Jones et al., 2008*). Second, the earlier used stimuli were deformable and presumably less rough (although they were not formally defined like our stimuli) than our solid (cause mounted in a frame) stimuli. Our results indicate that the effects of odor found for relatively smooth, deformable objects may not generalize to solid, rougher objects.

Summarizing, in contrast with previous studies that observed crossmodal interaction between olfaction and the tactile perception of textiles, skin and hair, the present study showed no effect of trigeminal and olfactory stimulation on tactile perception of the roughness of solid material. The absence of this effect may be of interest for the producers of cleaning products, since it implies that it will probably not be possible to influence the perceived effectiveness of these products through the addition of particular odorants.

## ACKNOWLEDGEMENTS

The authors thank Dr. Edwin Gelinck (TNO) for the microscopic examination of the surface roughness of the sandpaper samples. We also thank Dr. Stella Donker for her help with the recruitment of participants and for providing us with her lab facilities.

### Funding

This work was funded by TNO. The funders had no role in study design, data collection and analysis, decision to publish, or preparation of the manuscript.

### Competing Interests

All authors are employees of TNO Human Factors.

## Author Contributions

- Lara A. Koijck conceived and designed the experiments, performed the experiments, analyzed the data, contributed reagents/materials/analysis tools, wrote the paper, prepared figures and/or tables, reviewed drafts of the paper.
- Alexander Toet conceived and designed the experiments, performed the experiments, contributed reagents/materials/analysis tools, wrote the paper, prepared figures and/or tables, reviewed drafts of the paper.
- Jan B.F. Van Erp conceived and designed the experiments, analyzed the data, wrote the paper, reviewed drafts of the paper.

## Human Ethics

The following information was supplied relating to ethical approvals (i.e., approving body and any reference numbers):

1. The TNO Ethics Committee.

2. The protocol for the experiments was signed and approved by the TNO Ethics Committee, and thereafter approved and signed by the department head of TNO.

## Supplemental Information

Supplemental information for this article can be found online at http://dx.doi.org/10.7717/peerj.955#supplemental-information.

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
