# Peer review of "Tactile roughness perception in the presence of olfactory and trigeminal stimulants"

_PeerJ, doi:10.7717/peerj.955_

## Round 0.1 · original submission · Major Revisions

Both authors make specific suggestions to validate the methodology that I think are critical to resubmission. Please follow those. Both reviewers are skeptical that you would have found a result given your design. Whereas PeerJ does not require a surprising result (and therefore your results are valid for publication since methodologically verified above), journal policy is to appropriately discuss the results and so you must address the issue of whether or not your experiment could have found an interaction, and why the present result is significant in that it does not find the tested interaction.

Reviewer 1 ·

Basic reporting

Previous studies show that odor stimuli affect the tactile perception of material properties in certain cases. Authors examines whether the roughness perception of sandpapers is biased by different odor stimuli that have different subjective pleasantness and perceptual intensity. Participants evaluated four different grits of sand papers with and without odor stimuli. Under odorless condition, subjective roughness magnitude was statistically different between tested sandpapers, consistent with previous studies. Under odor condition, the change of participants’ roughness estimation was not statistically different from condition where odor stimuli is absent. Authors concluded that this may be due to the low concentration of odor stimuli.

Experimental design

The description in Materials and Methods is fair enough for replication, except the rationale how authors determined the number of participants. I would suggest that authors conduct control experiment using known odor stimulus that can bias the tactile perception of material properties (e.g., a lemon odor).

Validity of the findings

Authors conduct statistical analysis for their data, however, they did not statistically meaningful data that support their hypothesis.

Additional comments

Tactile perception is biased by olfactory stimulation is an interesting viewpoint that authors challenge. Overall, the reviewer thinks that this manuscript is within the scope of the PeerJ and, in a revised form, will highly appeal to the Journal's readership.

Two novel aspects of this study will represent significant advances to the field. First, the authors show that PEA and alcohol odors do not bias the tactile perception of roughness. Second, they demonstrate that this unbiased perception happens if the odor stimuli is under perceptual threshold. These results can be more interesting if authors could find the same result with using known odor stimuli that bias tactile perception of other material properties. Such study will increase the understanding of tactile perception, as well as the contribution of odor stimuli in cognition. In this manuscript, authors' result remains preliminary and the effect of odor stimuli in tactile roughness perception remains intact.

·

Basic reporting

No comments.

Experimental design

What is presented is fine, but as noted in my general comments, the authors really set themselves up to not find interactions.

Validity of the findings

No comments.

Additional comments

Koijk et al. investigated whether roughness judgments of sandpaper were influenced by the presence of an odorant in the room in which the judgments were obtained. No significant associations were found, although some trends were reported that were consistent with the hypothesized biases.

Overall, the work was conducted soundly, and the results and interpretation have no issues. Probably the main issue is that one would not necessarily expect a crossmodal interaction or bias if there was no temporal association between the sandpaper assessment and the odor. That is, crossmodal associations often require the multiple sensory inputs to occur at very close to the same time. However, in the experiment reported, the odors were present continually; i.e., they were not time-locked to any specific stimulus. Thus, the authors likely rendered themselves to likely not find interactions from the outset. Having said that, this observation does not invalidate the study. However, temporal issues in crossmodal binding should be noted in the Discussion.

---

## Round 0.2 · accepted · Accept

One of the reviewers re-read your manuscript and weas happy with your modifications. Thank you for responding to the review process.

·

Basic reporting

No Comments.

Experimental design

No Comments.

Validity of the findings

No Comments.

Additional comments

The authors added an experiment in this resubmission, which was more likely to uncover odor-touch interactions- a worthwhile addition. Although no interactions were found, that is not to say that other odor-touch interactions might not exist. As the authors suggest, odor-touch interactions might be quite domain / stimulus-specific, which is a reasonable statement.